# Relative Risk of Death in Bulgarian Cancer Patients during the Initial Waves of the COVID-19 Pandemic

**DOI:** 10.3390/healthcare11182594

**Published:** 2023-09-20

**Authors:** Velizar Shivarov, Denitsa Grigorova, Angel Yordanov

**Affiliations:** 1Department of Experimental Research, Medical University Pleven, 5800 Pleven, Bulgaria; 2Department of Probability, Operations Research and Statistics, Faculty of Mathematics and Informatics, Sofia University, 1504 Sofia, Bulgaria; dgrigorova@fmi.uni-sofia.bg; 3Department of Gynaecologic Oncology, Medical University Pleven, 5800 Pleven, Bulgaria; angel.jordanov@gmail.com

**Keywords:** COVID-19, cancer, mortality, relative risk, Bulgaria

## Abstract

Background: The COVID-19 pandemic has led to millions of documented deaths worldwide, with diverse distribution among countries. Surprisingly, Bulgaria, a middle-income European Union member state, ranked highest in COVID-19 mortality. This study aims to assess whether Bulgarian cancer patients experienced a higher relative risk (RR) of death compared to the general Bulgarian population during the pandemic. Materials and Methods: Data from the Bulgarian National Statistical Institute and the Bulgarian National Cancer Registry were analyzed to estimate monthly RR of death in cancer patients compared to the general population before and during the first two years of the pandemic. The impact of the COVID-19 waves and predominant SARS-CoV-2 variants on RR was evaluated on various cancer types and age groups using a multiple linear regression approach. Results: During the COVID-19 waves, both the general population and cancer patients experienced a significant increase in mortality rates. Surprisingly, the RR of death in cancer patients was lower during pandemic waves. The results from the statistical modeling revealed a significant association between the COVID-19 waves and reduced RR for all cancer patients. Notably, the effect was more pronounced during waves associated with the Alpha and Delta variants. The results also showed varying impacts of the COVID-19 waves on RR when we analyzed subsamples of data grouped depending on the cancer type, age and sex. Conclusions: Despite increased overall mortality in Bulgarian cancer patients during the pandemic, the RR of death was lower compared to the Bulgarian general population, indicating that protective measures were relatively effective in this vulnerable group. This study underscores the importance of implementing and encouraging preventive measures, especially in cancer patients, to mitigate the impact of future viral pandemics and reduce excess mortality.

## 1. Introduction

As of the end of May 2023 the COVID-19 pandemic led to almost 7,000,000 documented deaths worldwide (https://covid19.who.int/, accessed on 2 June 2023) with diverse distribution among countries. It was initially perceived that at an individual level mortality would be highly dependent on socio-demographic factors [1]. Therefore, surprisingly, the top positions in the ranking of the highest mortality are for two European Union (EU) member states, namely Bulgaria and Hungary, with estimated COVID-19 deaths per 100,000 inhabitants of 550.17 and 504.76, respectively (https://coronavirus.jhu.edu/data/mortality, accessed on 2 September 2023). Bulgaria was also shown to be one of the countries with highest number of excess death cases during the first two years of COVID-19 pandemic with 647.3 excess deaths per 100,000 inhabitants [2]. This high mortality rate in middle-income countries can be explained by a number of factors such as the general health status and age structure of the population, the overall level of development of the healthcare system in the country, acceptance of the general protective measures such as social distancing, mask wearing and vaccination practiced by the population, etc. [3]. On the other hand, the quality of the healthcare systems is highly dependent on the amount of healthcare expenditures, which may significantly influence the overall mortality in the country [4,5,6]. For example Bulgaria is the country with the lowest healthcare expenditures in the EU and with the shortest life expectancy in the pre-pandemic years [7]. Lower healthcare expenditures affect more severely vulnerable patient groups such as elderly and chronically ill people including cancer patients. Since 1998, Bulgaria has mixed public–private coverage of health care expenditures [8]. The main contributor to the health care costs coverage in the country is the National Health Insurance Fund (NHIF) which provides financial coverage based on the solidarity principle. NHIF also covers the costs for newly launched and expensive anti-cancer treatments. According to some estimates Bulgaria has a relatively high proportion of out-of-pocket payments for health care [8]. However, for more than 15 years Bulgaria has been also the country with lowest expenditures for oncology care in the EU [9,10]. According to the 2018 estimate of cancer care costs in EU, the direct healthcare expenditure on cancer care in Bulgaria was EUR 45 per capita [10]. This amount was slightly higher than the identical measure in Romania, which was EUR 36 per capita but more than four-fold lower than the Europe average expenditure of EUR 195 per capita [10]. On the other hand, based on the report by the European Cancer Information System (ECIS) (https://ecis.jrc.ec.europa.eu/, accessed on 10 September 2023) the age-adjusted cancer incidence and mortality in Bulgaria in 2020 were 458.0 and 258.4 per 100,000 inhabitants, respectively. These rates are lower than the European average but the same report showed that the 5 year relative survival of cancer patients is between 5% and 15% lower than the European average depending on sex and age group. Therefore, given the fact that cancer patients have been shown to be at increased risk for poor outcomes from COVID-19 [11,12] we questioned whether Bulgarian cancer patients had higher mortality than the general Bulgarian population. From a healthcare management perspective this question is of particular importance as its answer would provide the basis for the analysis of the effectiveness of the overall management and the targeted protective measures in cancer patients during the COVID-19 pandemic in a country with an overall high mortality rate during the pandemic.

To address this question, we analyzed the dynamics in the monthly relative risk (RR) of death in Bulgarian cancer patients immediately before and during the first two years of the COVID-19 pandemic. For the purpose of those analyses, we used publicly available data from the Bulgarian National Statistical Institute (NSI) and the Bulgarian National Cancer Registry (BNCR).

## 2. Materials and Methods

### 2.1. Data Sources

We obtained data regarding monthly and annual mortality for both sexes and all ages in Bulgaria between 1 January 2016 and 31 December 2021 from the data portal of the Bulgarian National Statistical Institute (https://infostat.nsi.bg/infostat/pages/external/login.jsf) (accessed on 1 February 2023). Monthly aggregate data regarding the newly diagnosed and dead cancer patients were obtained from the Bulgarian National Cancer Registry (BNCR) (https://www.sbaloncology.bg, accessed on 2 September 2023) covering the abovementioned period (last updated 1 February 2023). Data from the BNCR were stratified by age at diagnosis, sex and major entities. As BNCR was not able to provide the age at death or last follow-up we defined overlapping age groups at diagnosis as: all ages (designated as 0+), older than 19 years (designated as 19+), older than 39 years (designated as 19+) and older than 59 years (designated as 59+). The main entities groups were defined based on ICO-0-2 codes as follows: colorectal cancer (CRC) (C18–C21); acute myeloid leukemia (AML) (C92.0, C92.2–C92.9, C93.0, C93.2–C93.9, C94.2); lung cancer (LC) (C34); multiple myeloma (MM) (C90); ovarian cancer (OC) (C56); breast cancer (BC) (C50); prostate cancer (PC) (C61); endometrial cancer (EC) (C54.1); cervical cancer (CC) (C53); malignant lymphoma (ML) (C81–C85, C88.0, C91.1–C91.9); and chronic myeloid leukemia (CML) (C92.1).

### 2.2. Data Analyses

Monthly mortality for the general population (per 1000 people) per age and sex group was directly obtained from the NSI data portal. Monthly mortality in cancer patients required additional manipulations of the raw data provided by the BNCR which consist of the raw counts of registered cases and the number of deaths per month for each gender, age group and ICO-O-2 code described above. We started with the calculation of the average number of the registered patients in each patients’ subgroup formed by the cancer type, sex and age. Subsequently, the number of death cases per month was divided by the average number of the registered patients per group, adjusted for the length of the month and year (in days) and multiplied by 1000 to obtain the monthly mortality in the different cancer patients’ groups. The relative risk of death in every cancer patients’ group compared to the general population for a given month was calculated as the ratio of the proportions of the death cases (monthly mortality divided by the total number of subjects) in both cohorts of individuals (given cancer patient group and general population) [13]. For the latter calculation we subtracted the number of cancer patients from the total number of subjects in the general population and the number of deaths among cancer patients from the total number of deaths in the general population.

We modeled the relative risk of death in cancer patients compared to the general population using multiple linear regressions. The predictors are the presence or absence of a COVID-19 wave, the predominant SARS-CoV-2 variant in Bulgaria and the month of the year, all these variables were considered as categorical. This approach allows for analysis of the seasonality in the data assuming the month of the year as a fixed effect [14]. Information regarding COVID-19 waves and predominant strains were obtained from a previous publication [15]. Assumptions of the model were verified by analysis of the normal distribution of the residuals using a quantile–quantile (Q–Q) plot and Shapiro–Wilk test (Appendix A). All statistical analyses were performed using R for Windows (v. 4.3.0). The significance level was set to 5%. For the graphical illustration of the results from the analyses we used the packages *forestmodel* and *ggpubr*.

## 3. Results

We initially estimated and plotted the monthly death rates in Bulgaria for the general population and for the registered cancer patients merged and split by gender (Figure 1). As shown on Figure 1 the second, third and fourth COVID-19 waves in the second half of 2020 and during 2021 were associated with a dramatic increase in the mortality rates for both the general and the cancer patients’ population. Our main question, however, was if the effect of the COVID-19 waves is associated with proportionally equal or even increased mortality among cancer patients compared to the mortality in the general population. Surprisingly, the RR for death in cancer patients was lower during pandemic waves (Figure 1).

We focused on obtaining statistical evidence that the decrease in the RR of death in cancer patients was associated with COVID-19 pandemic waves when taking into account the seasonal variations in general mortality. For this purpose, we included the month of the year as a controlling factor variable in all models. The results from the model fitted to the data on all cancer patients showed that the presence of a wave was statistically significantly associated with a reduction in the RR (*p*-value = 0.022) and the effect was also significant for Alpha and Delta variant-associated waves (Figure 2). Identical analyses were performed separately for males and females for each age group. Of note, the presence of Alpha and Delta waves were almost invariably associated with a significant decrease in the RR (Figure 2).

We further extended our analyses to different cancer entities including only adult patients. We initially focused on CRC and LC patients above 19 years at diagnosis and observed a similar increase in mortality rate in CRC and LC of both sexes during COVID-19 albeit without an increase in the RR during the same period (Figure 3). The results from the statistical analysis showed that for male LC patients the presence of a COVID-19 wave was not associated with any difference in RR, however, the presence of any predominant variant was associated with lower RR. On the other hand, in female LC patients only the Delta variant was associated with a statistically significant decrease in RR. Regarding CRC patients, predominance of the Delta variant was associated with a decreased RR in both male and female CRC patients (Figure 4). In female CRC patients the predominance of the Alpha variant was also associated with a lower RR (Figure 4).

The sex-specific differences found in the RR for CRC patients prompted us to extend our analyses to reproductive system cancers in adult patients. The BC and EC female patients had a very prominent increase in mortality rate during COVID-19 waves but the effect was less evident for the CC and OC patients (Figure 5). The visual inspection of the dynamics of RR for female reproductive system cancers suggested that there might be some decrease during the COVID-19 waves (Figure 5).

The results, however, showed associations of the COVID-19 waves with RR in such female patients, i.e., the BC patients had a significantly lower RR during Delta variant-associated waves, whereas the same was true for OC patients during Alpha variant-associated waves (Figure 6). The only male-specific cancer type that we analyzed was PC (Figure 5). PC patients showed an increased mortality rate and decreased RR during COVID-19 waves (Figure 5). Notably, any type of predominant variant was associated with a statistically significant lower RR in PC patients (Figure 6).

The last group of patients which we studied was the group of liquid cancers (hematological malignancies). We defined several groups with different lineage origins and distinct clinical courses. We observed an increase in the mortality rate in lymphoma patients during the COVID-19 waves and minor effect in MM patients whereas for CML and AML there was no obvious effect on mortality rate (Figure 7).

Statistical analyses confirmed the observation from the graphics that the relative risk is not affected by the presence of a COVID-19 wave (Figure 8). There was only one statistically significant association: predominance of the Alpha variant was associated with lower RR in AML patients.

## 4. Discussion

Bulgaria had one of the leading positions in the ranking of the highest mortality from COVID-19 worldwide. We investigated if the mortality in one subgroup of the general Bulgarian population, namely the cancer patients, was proportional to the mortality in the general population before and during the first two years of COVID-19 (the period studied is from the beginning of 2016 till the end of 2022). In this study we analyzed data from the Bulgarian National Statistical Institute and the Bulgarian National Cancer Registry in order to estimate the monthly RR of death in cancer patients compared to the Bulgarian general population. Using multiple linear regression models for different subsamples of cancer patients depending on their cancer type, age and sex, we estimated the effects of the presence of a COVID-19 wave and the predominant SARS-CoV-2 variant in Bulgaria, controlling for the seasonal effect on mortality.

Mortality in the general population has a clear seasonality pattern with significant increase during the cold months of the year [16]. This might be due to variation in physiological parameters such as blood pressure, making susceptible populations more likely to experience cardiovascular fatalities [17]. On the other hand, seasonal variations in the immune system and more close contacts during the cold months increase the risk of severe and fatal infectious respiratory disease such as flu [18,19]. With the surge of the COVID-19 pandemic in the winter and early spring of 2020 there was initial hope that as a typical respiratory infection the pandemic may have a seasonal course with rapid self-limitation after the cold months [20]. However, SARS-CoV-2 transmission proved to be independent of climatic factors and the pandemic waves were largely dependent on the dynamics of population-based preventive measures, biological properties of the predominant and newly emerging virus variants and the penetration of specific prophylaxis through vaccination since early 2020. On the other hand, not surprisingly, during the early pandemic and with the spread of the disease it became evident that the risk of severe COVID-19 and a lethal outcome depends predominantly on the patient’s general health status [21]. Therefore, patients with acquired immune deficiencies such as cancer patients were shown to be at increased risk for death from COVID-19 [22]. Obviously, the overall mortality in cancer patients in any country during the pandemic would have been dependent on the general measures implemented to mitigate the risk of SARS-CoV-2 spread but also on targeted measures to protect and manage cancer patients with the COVID-19 disease. Additionally, comparison of mortality rates in the general population and cancer patients would provide an estimate of the effectiveness of the general measures and the focused approaches to vulnerable cohorts of patients.

We focused our study on Bulgarian cancer patients with the main goal to evaluate the dynamics of the relative risk of death among them before and during the first two years of the COVID-19 pandemic. Bulgaria was unique case for the management of the COVID-19 pandemic with suppression of the initial spread and waves with very restrictive measures including complete lockdown in March–May 2020 [23]. However, subsequent poor management and control of general preventive measures and vaccination hesitancy led to high mortality rates with subsequent waves [24,25]. This increase in in mortality affected both the general population and cancer patients in Bulgaria as evidenced by our descriptive analyses of monthly mortality during the period 2020–2021.

Based on that and the limited resources for oncology care in the country we questioned whether the relative risk of death in cancer patients was identical or even higher than in the general population. Surprisingly, we observed a statistically significant lower RR of death in the cancer patients for all age groups and for both sexes during the COVID-19 waves in the country. Indeed, the RR of death in cancer patients was lower in the winter months even before the pandemic. This suggested that even though there was excess mortality in cancer patients during those periods it was not proportionate to the mortality in the general population. This observation directly suggests that cancer patients were more protected than the general population. There is no reliable data source which can provide information regarding the incidence of COVID-19 in cancer patients. Lower incidence of COVID-19 in those patients because of stricter adherence to general preventive measures may have accounted for a lower RR of death in 2020. On the other hand, the more rapid dissemination of vaccination and less vaccine hesitancy among cancer patients in 2021 might have contributed to the prevention of COVID-19 deaths in 2021. In support of that hypothesis we observed a tendency for a lower relative risk of death in cancer patients during the Delta variant-dominated waves in the second half of 2021 in comparison to Alpha variant-dominated ones in the first half of 2021.

Expectedly, the effect of the COVID-19 waves on mortality differed between cancer types. It was most pronounced in male lung cancer patients suggesting that those patients were probably better protected during the waves. Less aggressive cancer such as colorectal cancer was associated with lower relative risk of death during the waves. Breast and ovarian cancers showed a similar pattern of relative risk reduction especially with later waves, which was more pronounced for BC patients. One cannot exclude the possibility that long-term usage of selective estrogen receptors (SERMs) has played a protective role in BC survivors suffering from COVID-19 [26,27,28]. On the other hand, endometrial and cervical cancers did not show a reduced RR of death during waves. This may be due to several reasons. Firstly, endometrial cancers are usually less aggressive with more long-term survivors among mid- and advanced aged women who tend to have behaviors similar to the general population. Secondly, cervical cancer in Bulgaria is more widely spread among younger women from ethnic minorities (Roma communities) with significantly lower vaccination coverage against traditional pathogens and human papillomavirus (HPV) [29,30,31]. These minorities in general have low vaccination coverage and during the pandemics had the slowest vaccination rates in Europe due to a number of factors [32].

Interestingly, males with prostate cancer who oftentimes require chronic long-term hormonal therapy had a significantly lower relative risk of death during the COVID-19 waves. Indeed it was proposed that androgens may play a role in COVID-19 pathogenesis through the androgen-regulated transmembrane protease, serine 2 (TMPRSS2), which also plays a critical role in SARS-CoV-2-host cell membrane fusion [33]. Androgen deprivation therapy (ADT) could down-regulate TMPRSS2 transcription in the airways and in the lungs, thereby mitigating the severity of a SARS-CoV-2 infection [34]. A number of studies compared clinical outcomes in COVID-19 PC patients treated with ADT versus patients not receiving ADT therapy [35,36,37,38]. The reported data is conflicting with some studies reporting a protective role of ADT [36,37], while others did not identify any statistical difference between the two groups of PC patients [35,38,39]. Those reports compared ADT-treated with non-ADT-treated PC patients while our study compared PC patients with the general male population. Provided that the vast majority of PC patients would have received some form of ADT therapy one can speculate that our data suggest some role protective role against death from COVID-19 in PC patients.

In general, patients with hematological malignancies have the highest levels of immune suppression because of the nature of the diseases directly affecting the immune system as well as because of the usage of highly myelosuppressive therapies. We analyzed four groups of patients with hematological malignancies as they differ by the natural course of the disease and available therapeutic approaches (patients with CML, ML, MM and AML). CML patients, for example, showed a similar pattern to the one observed for endometrial cancer patients without any effect of the COVID-19 waves on the RR of death. This is because CML patients have a near-normal quality of life on chronic therapy with tyrosine kinase inhibitors [40] and are more prone to follow the behavioral patterns of the general population rather than those of the cancer patients. Patients with mature lymphoid malignancies such as ML who also tend to have a more chronic course of the underlying malignancies also showed only a minor insignificant trend towards lower a RR of death during the COVID-19 waves. MM patients who are more severely immunocompromised because of continuous long-term exposure to B cell-depleting therapy had detectable but statistically insignificant reduction of the RR of death during the Alpha variant wave. Lack of a significant reduction of the relative risk in ML and MM patients during the late 2021 waves might also be explained by the poor immunogenicity to vaccination in those patients [41], which might be overcome with subsequent booster vaccinations [42]. Finally, AML patients who in general have a very aggressive course with short overall survival in Bulgaria [43] had a significantly lower RR of death during the late 2020 waves but not during the waves of 2021. This suggests that the general protective measures led to a short-term benefit in those patients, but this effect was short-lived probably because such patients cannot have infrequent healthcare encounters for more prolonged periods of time.

Our study has some limitations based on the type of source data obtained. We only had aggregate data and no individual data available, such as actual age at death, the cause of death or vaccination status of patients. We compensated for the inability to perform an age-adjusted calculation of mortality and relative risk by analyzing overlapping groups based on their age at diagnosis. One can also speculate that delayed diagnosis and registry reporting of new cases might have affected the overall number of cancer cases during that period. Delayed diagnosis and reporting might be a real phenomenon, however, we analyzed a relatively long period and used an average number of registered cancer patients over the year which would compensate partly for delayed registration. Besides, all cases registered based on death certificates are captured in a timely manner and therefore the chance of under-estimation of the relative risk is avoided. Of course, if we had individual data available from cancer patients with their vaccination status and cause of death we could have directly assessed the effect of vaccination on COVID-19 and other causes mortality but this was obviously impossible and remained beyond the scope of this work.

The study raises the interesting question of whether any specific protective measures were more efficient among cancer patients. As described above this question could not be addressed directly as we did not have specific data regarding the adherence of cancer patients to specific measures. It is rational to speculate that as the relative risk of death was lower with the later COVID-19 waves it was a combination of measures that contributed to this outcome. Firstly, by the second half of 2020 the vaccination coverage might have been higher among cancer patients and therefore the disease would have been less frequent and less severe in that group. Secondly, cancer patients might have more strictly followed the principles of personal protection than the people in the general population even when the latter had poorer adherence to those measures, thereby reducing the overall number of cases and deaths. Finally, cancer patients might have been hospitalized earlier than otherwise healthy individuals, which could also contribute to a lower relative risk of death. Assessment of all these explanations requires more granular individual level data for death cases in the general population and the group of cancer patients.

## 5. Conclusions

In conclusion, our comprehensive analysis highlights that cancer patients in Bulgaria faced increased overall mortality during the initial two years of the COVID-19 pandemic. However, the intriguing finding is that the relative risk of death in cancer patients was surprisingly lower compared to the general population and the same analyses applied to different subgroups categorized by cancer type, age and sex showed similar findings for some of the cohorts considered. This observation strongly suggests that a substantial proportion of Bulgarian cancer patients diligently followed and adhered to general protective measures, including vaccination.

Regrettably, this lower RR in cancer patients implies that a significant portion of the excessive mortality in the Bulgarian general population could have been mitigated if similar protective measures were widely embraced and implemented. The results underscore the critical importance of public health initiatives that prioritize the protection of vulnerable groups, such as cancer patients, during future health crises and pandemics.

Moving forward, targeted efforts and tailored interventions should be put in place to ensure that high-risk populations receive adequate support and protection during any public health emergency. This includes not only emphasizing general preventive measures but also streamlining vaccination strategies and promoting healthcare access for vulnerable individuals. By proactively safeguarding those at higher risk, we can strive to reduce overall mortality rates and enhance the resilience of our healthcare systems in the face of future challenges.

## Figures and Tables

**Figure 1 healthcare-11-02594-f001:**
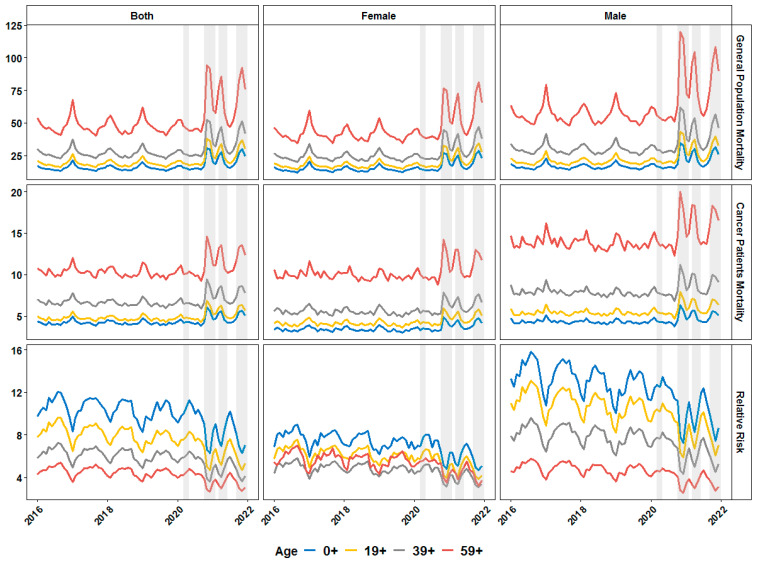
Monthly dynamics in mortality rate (per 1000 individuals) for general population (first row of graphics) and cancer patients (second row) and relative risk of death in cancer patients before and during the COVID-19 pandemic in Bulgaria (first column—both genders merged) and split by gender (second and third column) for different age groups. Grey-shaded areas denote the periods of documented surges in the incidence of SARS-CoV-2 infections in the country in 2020 and 2021.

**Figure 2 healthcare-11-02594-f002:**
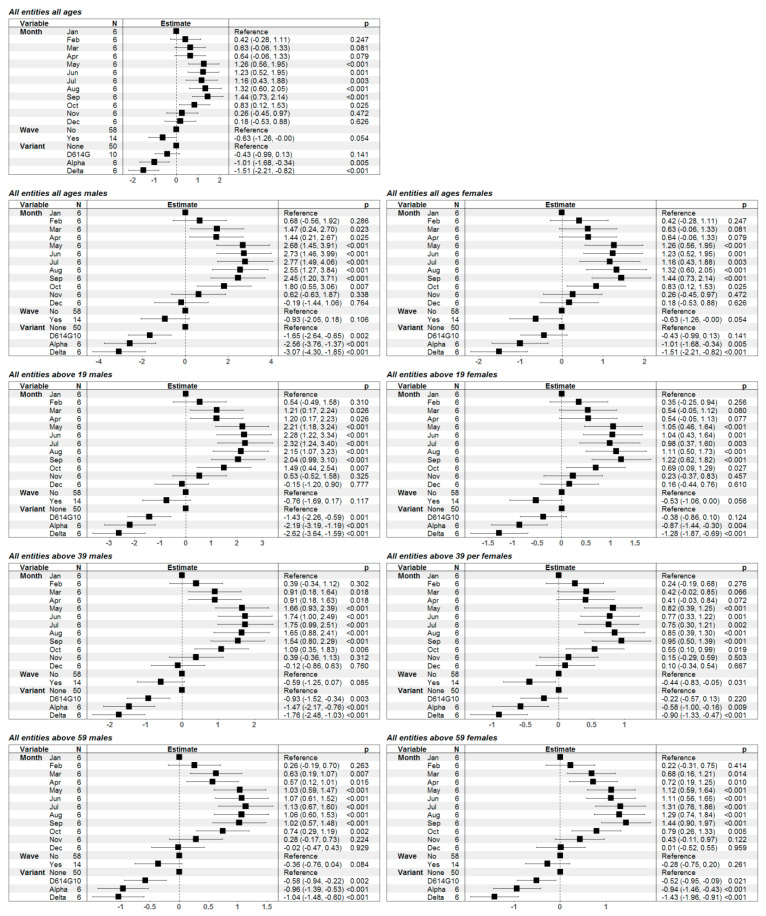
Graphical illustration and numerical presentation of the results from the multiple linear regression models for the risk ratio of death in cancer patients for different age and gender groups. In the first column of each separate table we present the predictor variables in the model (month of the year, presence or absence of a COVID-19 wave and the predominant SARS-CoV-2 variant); the second—graphical illustration of the estimates of the model parameters with 95% confidence intervals; the third—numerical presentation of the estimates with 95% confidence intervals in brackets along with the *p*-values for assessment of the statistical significance of the variables.

**Figure 3 healthcare-11-02594-f003:**
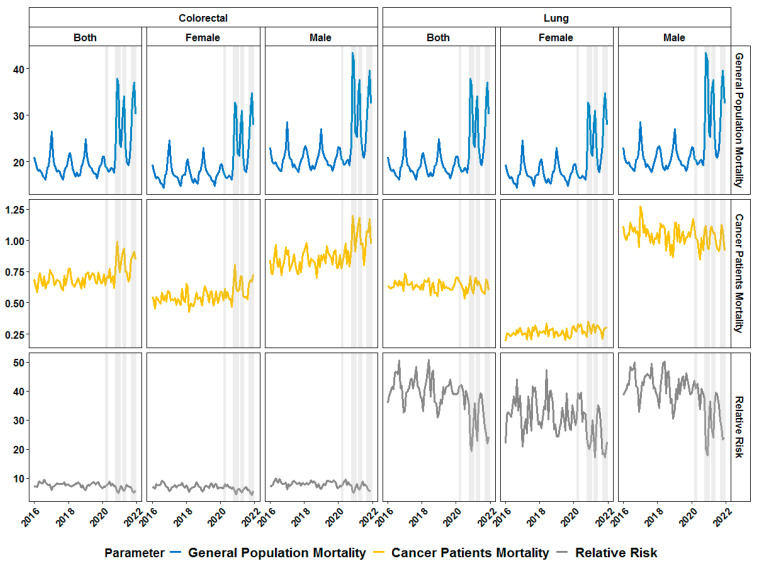
Monthly dynamics in mortality rate (per 1000 individuals) for general population (first row of graphics) and cancer patients (second row) and relative risk of death (third row) in patients with colorectal (left-hand block of graphics) and lung cancer (right-hand block of graphics) before and during the COVID-19 pandemic in Bulgaria for both genders, merged (first column in the each block) and split by gender (second and third column in each block). Grey-shaded areas denote the periods of documented surges in the incidence of SARS-CoV-2 infections in the country in 2020 and 2021.

**Figure 4 healthcare-11-02594-f004:**
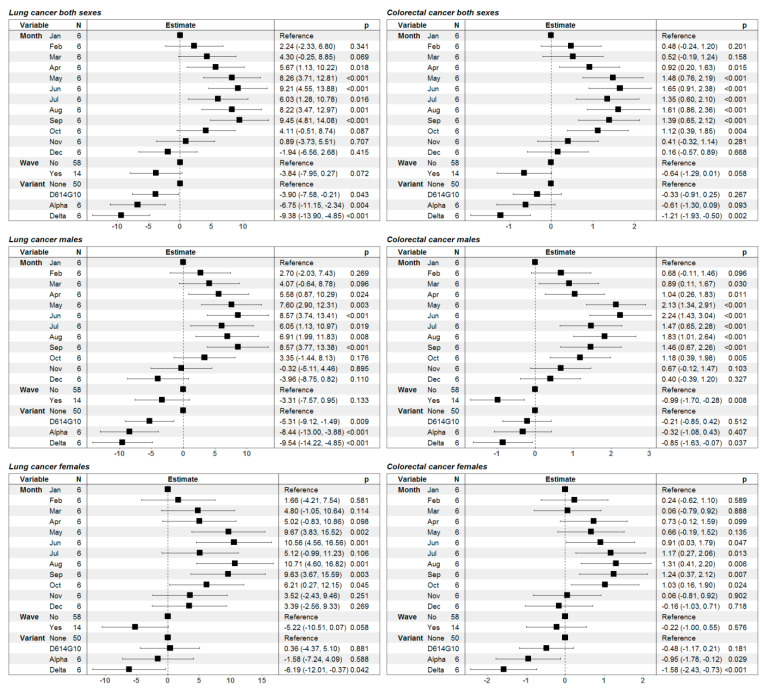
Graphical illustration and numerical presentation summarizing the results from the multiple linear regression models for the risk ratio of death in lung cancer patients (left-hand side) and colorectal cancer patients (right-hand side) for both genders, merged (first row of tables) and split by gender (second and third row of tables). In the first column of each separate table we present the predictor variables in the model (month of the year, presence or absence of a COVID-19 wave and the predominant SARS-CoV-2 variant); the second—graphical illustration of the estimates of the model parameters with 95% confidence intervals; the third—numerical presentation of the estimates with 95% confidence intervals in brackets along with the *p*-values for assessment of the statistical significance of the variables.

**Figure 5 healthcare-11-02594-f005:**
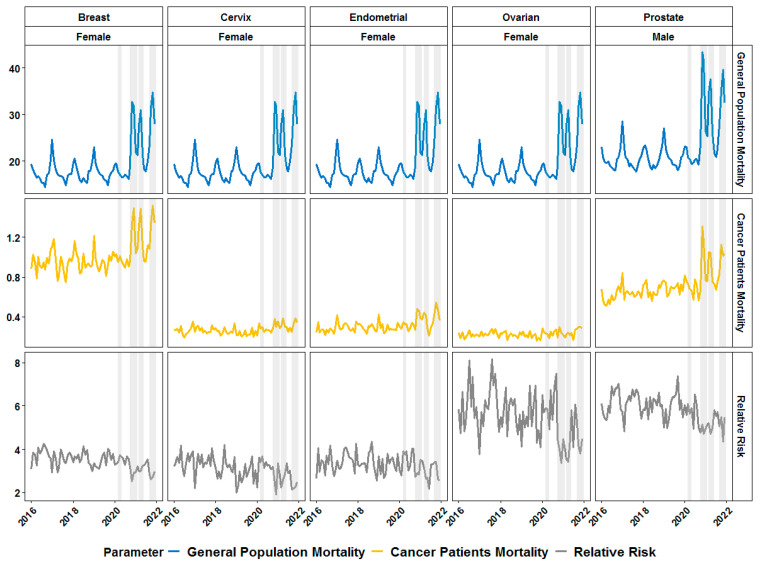
Monthly dynamics in mortality rate (per 1000 individuals) for general population (first row of graphics) and patients with genital cancer (second row) and relative risk of death (third row) in female patients with breast cancer (first column), cervix cancer (second column), endometrial cancer (third column), ovarian cancer (fourth column) and male prostate cancer patients (fifth column) before and during the COVID-19 pandemic in Bulgaria. Grey-shaded areas denote the periods of documented surges in the incidence of SARS-CoV-2 infections in the country in 2020 and 2021.

**Figure 6 healthcare-11-02594-f006:**
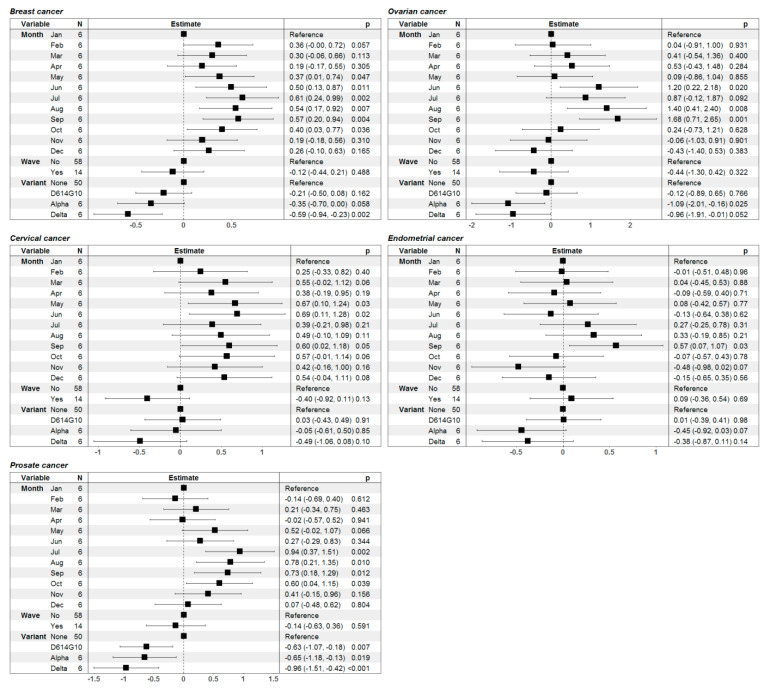
Graphical illustration and numerical presentation summarizing the results from the multiple linear regression models for the risk ratio of death in patients with genital cancer (breast, ovarian, cervical, endometrial and prostate cancer). In the first column of each separate table we present the predictor variables in the model (month of the year, presence or absence of COVID-19 wave and the predominant SARS-CoV-2 variant); the second—graphical illustration of the estimates of the model parameters with 95% confidence intervals; the third—numerical presentation of the estimates with 95% confidence intervals in brackets along with the *p*-values for assessment of the statistical significance of the variables.

**Figure 7 healthcare-11-02594-f007:**
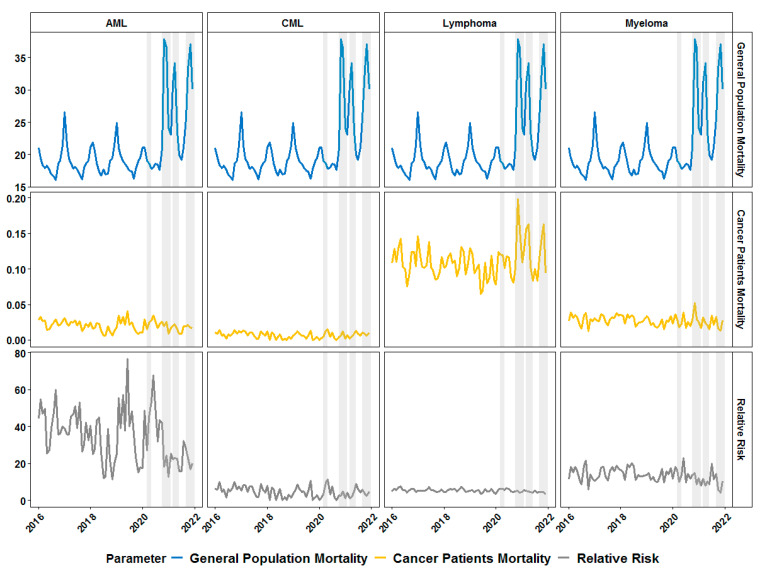
Monthly dynamics in mortality rate (per 1000 individuals) for general population (first row of graphics) and patients with hematological malignancies (second row) and relative risk of death (third row) in patients with acute myeloid leukemia (AML, first column), chronic myeloid leukemia (CML, second column), lymphoma (third column) and myeloma (fourth column) before and during the COVID-19 pandemic in Bulgaria. Grey-shaded areas denote the periods of documented surges in the incidence of SARS-CoV-2 infections in the country in 2020 and 2021.

**Figure 8 healthcare-11-02594-f008:**
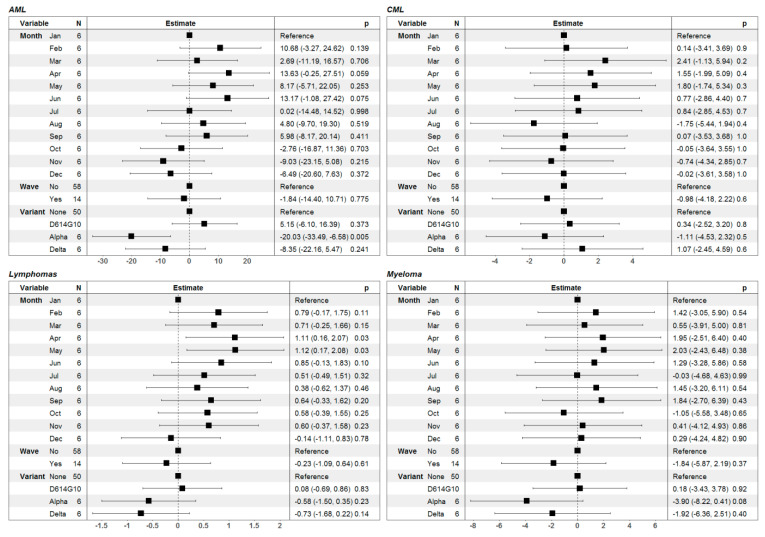
Graphical illustration and numerical presentation summarizing the results from the multiple linear regression models for the risk ratio of death in patients with hematological malignancies (acute myeloid leukemia (AML), chronic myeloid leukemia (CML), lymphomas and myeloma cancer). In the first column of each separate table we present the predictor variables in the model (month of the year, presence or absence of COVID-19 wave and the predominant SARS-CoV-2 variant); the second—graphical illustration of the estimates of the model parameters with 95% confidence intervals; the third—numerical presentation of the estimates with 95% confidence intervals in brackets along with the *p*-values for assessment of the statistical significance of the variables.

## Data Availability

Data used in the study can be obtained for free from the National Statistical Institute, Bulgaria, and the Bulgarian National Cancer Registry as described in the main text. Data and statistical analysis scripts are also available from the authors upon reasonable request.

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
