# Peer review of "Relative Risk of Death in Bulgarian Cancer Patients during the Initial Waves of the COVID-19 Pandemic"

_healthcare, 2023, doi:10.3390/healthcare11182594_

Round 1

Reviewer 1 Report

The topic is interesting and quite well developed.

I have some points to make:

- Vaccination rate (is it possible to have data of vaccinated patients vs.

General population?)

- the authors comment on the lower mortality risk in patients with prostate neoplasia, it would be useful to bring back into the discussion the hormonal aspect in breast cancer as well (e.g. PMID: 33244989, PMID: 33593260)

- in 2022 there was omicron as the predominant variant: can variant considerations be made?

None to add

Author Response

Reviewer: The topic is interesting and quite well developed.

Reply: Thank you for this evaluation.

Reviewer: I have some points to make:

- Vaccination rate (is it possible to have data of vaccinated patients vs.

General population?)

Reply: We agree that this would be an important piece of data which can be used to explain our findings unfortunately this data is not available for public use.

Reviewer: - the authors comment on the lower mortality risk in patients with prostate neoplasia, it would be useful to bring back into the discussion the hormonal aspect in breast cancer as well (e.g. PMID: 33244989, PMID: 33593260)

Reply: We appreciate that suggestion. We acknowledged this possibility by the following edit: “Breast and ovarian cancers showed similar pattern of relative risk reduction especially with later waves, which was more pronounced for BC patients. One cannot exclude the possibility that long-term usage of selective estrogen receptors (SERMs) has played a protective role in BC survivors suffering from COVID-190”

Reviewer: - in 2022 there was omicron as the predominant variant: can variant considerations be made?

Reply: Our analysis spans 2020-2021. At the time we applied for data access data for 2022 were not available for public use.

Reviewer 2 Report

Thank you very much for having the opportunity to review this paper. Some suggestions and questions are sent to the authors below:

1. Lines 53-54 says “Notably for more than 15 years Bulgaria has been also the country with lowest expenditures for oncology care in the EU”. The incidence of cancer in the Bulgarian population should be discussed by the authors. It is also suggested that the authors include information on expenditure on oncological care by person and compare this value with some other EU countries in order to support this statement.

2. Authors are encouraged to include a discussion of the health care system in Bulgaria in the Introduction section. Is it public or private funding for the general population? How is cancer treatment financed?  

3. Healthcare systems around the world were overwhelmed during the pandemic. Much is being discussed about its impact on the diagnosis of late-stage diseases such as cancer and other serious illnesses. Based on the authors' mention, I understand that the health system of Bulgaria has lower expenditures compared to EU countries. Authors are encouraged to present a discussion on the impact of the pandemic in the Bulgarian health system and consequences in cancer diagnosis. Is it possible that the number of new cancer cases could have been affected during the pandemic period?

4. Line 99 says “We need to note that the cancer mortality is included in the general population mortality”. If this form of calculation is standard in the field, please cite the bibliographical references. Although, it seems to me that this consideration by the authors is inappropriate and could be a source of bias in the results. Hypothetically, consider the following example: in a given period, there are a high number of deaths from colorectal cancer and lung cancer. On the other hand, the prostate cancer rate has stabilized during this time period. In this scenario, the RR of the general population will be increased compared with prostate cancer cases and it occurred by increasing other cancer cases. Based on this, I understand that it is not appropriate to include cancer mortality in general population mortality.

Authors are asked to exclude cancer mortality from the general mortality

of the population if there is no justification for the calculation made.

5. Considering the hypothesis that it is possible that many cases of cancer were not diagnosed during the pandemic, these persons were considered as the general population. Authors are encouraged to present and discuss this possibility as a limitation of the manuscript or as a possible explanation for the results obtained. This is very important since, if it is possible that many cancer cases were not reported due to the pandemic, this fact would contradict the results obtained, because the mortality rate among cancer patients would be much higher.

6. In addition to analyzing each cancer separately, authors are encouraged to evaluate a joint analysis that considers all cancer types together.

7. The vaccination, the lockdown and the seasons strongly influence the death rate of COVID, but it was not made not clear to me that the month would have a direct influence. What is the authors' justification for considering month and not considering vaccination, quarantine, and season as independent variables?

8. Why did the authors choose to consider age groups not exclusive? There are deaths being considered in all age groups. It is suggested that the authors consider the following age groups in the analyses presented: “0-18”, “19-38”, “39-48”, “59+”  (with a closed end).

9. Since all of the independent variables in the model are qualitative/categorical, why did the authors use the linear regression rather than variance analysis model? According to the specialized bibliography, when all independent variables are qualitative, the variance analysis model should be more appropriate (if the model assumptions were assessed satisfied).

10. Did the authors assess the interaction between the factors? Authors are encouraged to assess this effect if they have not already done it.

11. Have the assumptions of the model been verified? Please, include the normality test used in the manuscript. Is the independence assumption satisfied? Is it possible to assume that the number of deaths in one period is independent of the number of deaths in the previous period? At first glance, this assumption seems invalid due to the nature of the data. Please, clarify.

12. Delayed reporting is a major issue with COVID data in many countries. Has this problem occurred in Bulgaria? If so, how do the authors handle the data to address the issue?

13. Lines 256-257 says “Mortality in the general population has a clear seasonality pattern with significant increase during the cold months of the year”. It is noted that those dying from cancer also tend to be cyclical in nature. What justifies this behavior?

14. Lines 245-246 says “Bulgaria had the leading position in the ranking of highest mortality from COVID-19”. Is this information regarding the EU or global ranking? Please clarify.

15. Lines 329-331 says ““Provided that the vast majority of PC patients would have received some form of ADT therapy one can speculate that our data suggest some role protective role against death from COVID-19 in PC patients”. It is suggested that the authors remove this comment as it is speculative only.

During the reading of the manuscript, some English language mistakes were found (verbal concordances, spelling mistakes). It is recommended that the authors carefully review the whole text.

Author Response

Reviewer: Thank you very much for having the opportunity to review this paper. Some suggestions and questions are sent to the authors below:

Reviewer: 1. Lines 53-54 says “Notably for more than 15 years Bulgaria has been also the country with lowest expenditures for oncology care in the EU”. The incidence of cancer in the Bulgarian population should be discussed by the authors. It is also suggested that the authors include information on expenditure on oncological care by person and compare this value with some other EU countries in order to support this statement.

Reply: We appreciate that suggestion and added the following text: “According to the 2018 estimate of cancer care costs in European Union the direct healthcare expenditure on cancer care in Bulgaria was 45 Euro per capita. This amount was slightly higher than the identical measure in Romania, which was 36 Euro per capita but more than  four-fold lower than the Europe average expenditure of 195 Euro per capita. On the other hand based on the report by the European Cancer Information System (ECIS) (https://ecis.jrc.ec.europa.eu/) (last accessed 10-Sep-2023) the age-adjusted cancer incidence and mortality in Bulgaria in 2020 were 458.0 and 258.4 per 100000 inhabitants, respectively. These rates are lower than the European average but the same report showed that the 5-year relative survival of cancer patients is between 5% and 15% lower than the European average depeding on sex and age group. Therefore, given”

Reviewer: 2. Authors are encouraged to include a discussion of the health care system in Bulgaria in the Introduction section. Is it public or private funding for the general population? How is cancer treatment financed?

Reply: We added the following clarifying text per reviewer’s requirement: “Since 1998 Bulgaria has mixed public-private coverage of the health care expenditures (8). The main contributor to health care costs coverage in the country is the National Health Insurance Fund (NHIF) which provides financial coverage based on the solidarity principle. NHIF also covers the costs for newly launched and expensive anti-cancer treatment. According to some estimates Bulgaria has a relatively high proportion of out-of-pockets payments for health care (8). However,..”

Reviewer: 3. Healthcare systems around the world were overwhelmed during the pandemic. Much is being discussed about its impact on the diagnosis of late-stage diseases such as cancer and other serious illnesses. Based on the authors' mention, I understand that the health system of Bulgaria has lower expenditures compared to EU countries. Authors are encouraged to present a discussion on the impact of the pandemic in the Bulgarian health system and consequences in cancer diagnosis. Is it possible that the number of new cancer cases could have been affected during the pandemic period?

Reply: There was for sure delayed reporting in cancer registries. However, we covered a relatively long period of two years so after initial delay this must have been compensated. Besides, cancer cases registered based on death certificates are not significantly delayed. In other words, we capture accurately all deaths in cancer patients, therefore the risk for under-estimation of the RR was avoided.

Reviewer 4. Line 99 says “We need to note that the cancer mortality is included in the general population mortality”. If this form of calculation is standard in the field, please cite the bibliographical references. Although, it seems to me that this consideration by the authors is inappropriate and could be a source of bias in the results. Hypothetically, consider the following example: in a given period, there are a high number of deaths from colorectal cancer and lung cancer. On the other hand, the prostate cancer rate has stabilized during this time period. In this scenario, the RR of the general population will be increased compared with prostate cancer cases and it occurred by increasing other cancer cases. Based on this, I understand that it is not appropriate to include cancer mortality in general population mortality.

Authors are asked to exclude cancer mortality from the general mortality of the population if there is no justification for the calculation made.

Reply: We revised our methodology as requested by the reviewer and all models were recalculated and all figures redrawn. As you will find out there was minimal change in models estimates, which suggested that our original approach was also valid. We kept the revised figures to satisfy reviewer’s request and also revised the methodology section as follows:

“The relative risk of death in every cancer patients group compared to the general population for a given month was calculated as the ratio of the proportions of the death cases (monthly mortality divided by the total number of subjects) in both cohorts of individuals (given cancer patient group and general population). For the latter calculation we subtracted the number of cancer patients from the total number of subjects in the general population and the number of deaths among cancer patients from the total number of deaths in the general population.”

Reviewer 5. Considering the hypothesis that it is possible that many cases of cancer were not diagnosed during the pandemic, these persons were considered as the general population. Authors are encouraged to present and discuss this possibility as a limitation of the manuscript or as a possible explanation for the results obtained. This is very important since, if it is possible that many cancer cases were not reported due to the pandemic, this fact would contradict the results obtained, because the mortality rate among cancer patients would be much higher.

Reply: Please, see our reply above where we explained why the risk for under-estimation of the RR was avoided. We included a section regarding the limitations of the study as follows:

“Our study has some limitations based on the type of source data obtained. We had available only aggregate data and no individual data, such as actual age at death, cause of death or vaccination status of patients. We compensated the inability to perform age adjusted calculation of mortality and relative risk by analyzing overlapping groups based on the age at diagnosis. One can also speculate that delayed diagnosis and registry reporting of new cases might have affected the overall number of cancer cases during that period. Delayed diagnosis and reporting might be a true phenomenon however we analyzed a relatively long period and used average number of registered cancer patients over the year which would compensate partly to delayed registration. Besides, all cases registered based on death certificates are captured in a timely manner and therefore the risk of under-estimation of the relative risk is avoided. Of course if we had available individual data from cancer patients with their vaccination status and cause of death we could have directly assessed the effect of vaccination on COVID-19 and other causes mortality but this was obviously impossible and remained beyond the scope of this work.”

Reviewer: 6. In addition to analyzing each cancer separately, authors are encouraged to evaluate a joint analysis that considers all cancer types together.

Reply: The first analysis described in the Results section was performed to all cancer patients (first paragraph in Results section, Figures 1 and 2).

Joint analysis of all studied cancer types in terms of joint statistical modelling is not feasible in our case because of the big number of cancer types considered. More than two outcome variables are very rarely considered in a joint model in the applied statistical literature. These models are very sophisticated and usually there is no software implementation.

Reviewer: 7. The vaccination, the lockdown and the seasons strongly influence the death rate of COVID, but it was not made not clear to me that the month would have a direct influence. What is the authors' justification for considering month and not considering vaccination, quarantine, and season as independent variables?

Reply: All these covariates are not independent from presence of waves and therefore not included in our models. Our goal was to investigate how the seasonality in the relative risk of death for cancer patients changed during the pandemic. We have explained above why this approach to analyze seasonality in RR was adequate.

Reviewer: 8. Why did the authors choose to consider age groups not exclusive? There are deaths being considered in all age groups. It is suggested that the authors consider the following age groups in the analyses presented: “0-18”, “19-38”, “39-48”, “59+”  (with a closed end).

Reply: The cancer registry does not provide data regarding the actual age at death/time of data download therefore we had the patients grouped based on the age at diagnosis. Based on our approach one can make sure that the patients in a given age group are at a giver age or older. Using multiple overlapping age groups partially compensates for the lack of the data of exact age at death.

Reviewer: 9. Since all of the independent variables in the model are qualitative/categorical, why did the authors use the linear regression rather than variance analysis model? According to the specialized bibliography, when all independent variables are qualitative, the variance analysis model should be more appropriate (if the model assumptions were assessed satisfied).

Reply: We agree that the comparison of the risk ratios among the different groups formed by the different combinations of levels of the categorical variables is essentially analysis of variance. But multiple linear regressions is a class of models that incorporates the analysis of variance where the independent variables can be any mix of categorical and numerical variables. In the manuscript we wanted to present the results from the analysis not only as a conclusion about statistically significant presence of difference in risk ratios but also with some numbers from the linear regression outputs that are easy for interpretation and allow for quick quantitative comparison of the different levels of the predictor variables. As explained in the main text, infectious diseases mortality and general mortality show seasonal pattern. We used month of death as a fixed effect to model for underlying seasonality. Using this model we can estimate whether the covariates such as age, presence of waves and predominant virus strain actually modify the fixed effect of month on mortality (seasonality). This is a standard approach to analyze seasonal health data and we have referenced the major methodological guideline for this approach.

Reviewer: 10. Did the authors assess the interaction between the factors? Authors are encouraged to assess this effect if they have not already done it.

Reply: We assessed such possibility. Adding interaction terms in our models did not improve the  model. When we fitted models with interaction terms we encountered problems in the estimation procedure. We believe this is a signal of overfitting having too many parameters in the model.

Reviewer: 11. Have the assumptions of the model been verified? Please, include the normality test used in the manuscript. Is the independence assumption satisfied? Is it possible to assume that the number of deaths in one period is independent of the number of deaths in the previous period? At first glance, this assumption seems invalid due to the nature of the data. Please, clarify.

Reply: Yes. We provide Supplementary Figure 1 with a Q-Q plot of the residuals and Shapiro-Wilk test. The following text was added to the Methods section:

“Assumptions of the model were verified by analysis of the normal distribution of the residuals using quantile-quantile (Q-Q) plot and Shapiro-Wilk test (Supplementary Figure 1.).”

Including the month of the year as controlling variable is expected to account, at least partially if not entirely, the effect of the natural correlation of consecutive observations over time. We also tried models with correlated errors (for example ARMA(1,1)).

Reviewer: 12. Delayed reporting is a major issue with COVID data in many countries. Has this problem occurred in Bulgaria? If so, how do the authors handle the data to address the issue?

Reply: Please refer to the newly entered section regarding the limitations of the study which provides explanation to that.

Reviewer: 13. Lines 256-257 says “Mortality in the general population has a clear seasonality pattern with significant increase during the cold months of the year”. It is noted that those dying from cancer also tend to be cyclical in nature. What justifies this behavior?

Reply: The mortality in cancer patients follows identical seasonal pattern as the general population. This suggests that the risk factors for death affecting general population also affect or are applicable to cancer patients. However, the relative risk of death in cancer patients is lower during the cold season but remains stable over the warm months and higher than the general population.

Reviewer: 14. Lines 245-246 says “Bulgaria had the leading position in the ranking of highest mortality from COVID-19”. Is this information regarding the EU or global ranking? Please clarify.

Reply: We modified the sentence as follows: “Bulgaria had one of the leading position in the ranking of highest mortality from COVID-19 worldwide.”

Reviewer: 15. Lines 329-331 says ““Provided that the vast majority of PC patients would have received some form of ADT therapy one can speculate that our data suggest some role protective role against death from COVID-19 in PC patients”. It is suggested that the authors remove this comment as it is speculative only.

Reply: We opt to keep that sentence as Reviewer 1 requested that we discuss identical hypothesis for BC patients, which we did.

Reviewer 3 Report

Points of Positive Evaluation:

  1. Clear Objective and Significance: The study's objective is well-defined and significant, aiming to assess the impact of the COVID-19 pandemic on the relative risk of death for Bulgarian cancer patients compared to the general population. This topic holds importance in understanding the vulnerability of cancer patients during a pandemic.

  2. Relevant Background: The introduction provides relevant context by highlighting the global impact of the COVID-19 pandemic and its surprising effect on Bulgaria's mortality rate. This helps readers understand the need for the study and its potential implications.

  3. Comprehensive Data Sources: The study utilizes data from reputable sources, including the Bulgarian National Statistical Institute and the Bulgarian National Cancer Registry. This lends credibility to the analysis and findings.

  4. Methodological Clarity: The "Materials and Methods" section outlines the data analysis process clearly, specifying the use of monthly relative risk calculations and a multiple linear regression approach. This enhances the study's replicability and allows for potential validation of the results.

  5. Statistical Rigor: The study employs statistical modeling to assess the impact of COVID-19 waves and SARS-CoV-2 variants on relative risk for different cancer types and age groups. This approach enhances the depth of analysis and provides valuable insights.

  6. Contrasting Results: The unexpected finding that the relative risk of death for cancer patients was lower during pandemic waves adds a layer of complexity to the study. This finding challenges assumptions and prompts further investigation.

  7. Specific Findings: The study's findings are detailed and presented clearly, emphasizing the varying impacts of COVID-19 waves and the influence of specific variants on relative risk. This level of granularity contributes to a richer understanding of the subject matter.

  8. Practical Implications: The conclusion effectively relates the study's findings to real-world implications. It emphasizes the importance of preventive measures for vulnerable groups, like cancer patients, in future viral pandemics. This could guide public health policies.

Points of Negative Evaluation:

  1. Limited Discussion of Limitations: The study does not extensively address potential limitations, such as data quality or potential confounding factors. Addressing limitations could bolster the study's credibility and demonstrate a comprehensive analysis.

  2. Lack of External Validation: The study's results could have been strengthened by validating them with data from other countries or regions. This would provide a broader context and enhance the generalizability of findings.

  3. Incomplete Contextualization of Findings: While the conclusion suggests that protective measures were effective, it does not provide insight into the nature of these measures or their specific impact. Providing more context could enhance the study's implications.

  4. Scope of Variants: The study primarily focuses on the Alpha and Delta variants' influence on relative risk. A more comprehensive examination of other variants could have enriched the analysis further.

  5. Clarity on Overall Mortality: The study mentions increased overall mortality among Bulgarian cancer patients during the pandemic but does not delve into the specifics or possible causes for this increase. Further exploration of this aspect could offer additional insights.

  6. Absence of Ethical Considerations: The study does not address ethical considerations, such as patient consent or data privacy. Discussing these aspects is important in research involving human subjects.

Improve quality of english language.

Author Response

Points of Positive Evaluation:

Reply: Thanks for the positive evaluations of various aspects of our manuscript.

Points of Negative Evaluation:

Reviewer: 1. Limited Discussion of Limitations: The study does not extensively address potential limitations, such as data quality or potential confounding factors. Addressing limitations could bolster the study's credibility and demonstrate a comprehensive analysis.

Reply: We included a specific paragraph discussing the limitations of the study.

Reviewer: 2. Lack of External Validation: The study's results could have been strengthened by validating them with data from other countries or regions. This would provide a broader context and enhance the generalizability of findings.

Reply: We had such data available for Bulgaria and it was beyond the scope of this work to make generalizations for a number of countries. The goal of this work was to analyze if cancer patients were more or less severely affected by COVID-19 in terms of mortality.

Reviewer: Incomplete Contextualization of Findings: While the conclusion suggests that protective measures were effective, it does not provide insight into the nature of these measures or their specific impact. Providing more context could enhance the study's implications.

Reply: We added the following paragraph to the discussion section: “The study raises the interesting question if any specific protective measures were more efficient among cancer patients. As described above this question could not be addressed directly as we did not have specific data regarding the adherence of cancer patients to specific measures. It is rational to speculate that as the relative risk of death was lower with later COVID-19 waves it was the combination of measures that contributed to this outcome. Firstly, by the second half of 2020 the vaccination coverage might have been higher among cancer patients and therefore the disease would have been less frequent and less severe in that group. Secondly, cancer patients might have followed more strictly the principles of personal protection than the people in the general population even when the latter had poorer adherence to those measures thereby reducing the overall number of cases and deaths. Finally, cancer patients might have been hospitalized earlier than otherwise healthy individuals, which could also contribute to lower relative risk of death. Assessment of all these explanations requires more granular individual level data for deaths cases in the general population and the group of cancer patients.

Reviewer: Scope of Variants: The study primarily focuses on the Alpha and Delta variants' influence on relative risk. A more comprehensive examination of other variants could have enriched the analysis further.

Reply: As answered to similar comment by Reviewer 1 the study was limited to 2020-21 period and therefore subsequent dominant variants were not included in the analyses.

Reviewer: Clarity on Overall Mortality: The study mentions increased overall mortality among Bulgarian cancer patients during the pandemic but does not delve into the specifics or possible causes for this increase. Further exploration of this aspect could offer additional insights.

Reply: We modified discussion a bit to emphasize that increased mortality was due to COVID-19 spread in the general and cancer patients population as follows: “However, subsequent poor management and control of general preventive measures and vaccination hesitancy led to high mortality rates with subsequent waves (25, 26). This increase in in mortality affected both general population and cancer patients in Bulgaria as evidenced by our descriptive analyses of monthly mortality during the period 2020-2021.”

Reviewer: Absence of Ethical Considerations: The study does not address ethical considerations, such as patient consent or data privacy. Discussing these aspects is important in research involving human subjects.

Reply: We included specific statements in that regards as also requested by the editor as follows: Institutional Review Board Statement: This work is considered exempt from IRB review as it uses free publicly available aggregate statistical data.

Informed Consent Statement: Not applicable as this study uses free publicly available aggregate statistical data. There were no privacy issues to be considered in regards to the usage of such data.

Round 2

Reviewer 1 Report

None to add

Reviewer 2 Report

I would like to thank the authors for answering my questions, accepting the main suggestions and making the needed changes in the manuscript. I consider that the manuscript is ready to be published.

During the new reading of the manuscript, some English language mistakes were found.

Reviewer 3 Report

Agree. Accepted.